# Balanced gene dosage control rather than parental origin underpins genomic imprinting

Ariella Weinberg-Shukron[1,2,3], Raz Ben-Yair[1,3], Nozomi Takahashi[2], Marko Dunjić[1], Alon Shtrikman [1], Carol A. Edwards [2], Anne C. Ferguson-Smith [2] ✉ & Yonatan Stelzer [1] ✉

Mammalian parental imprinting represents an exquisite form of epigenetic control regulating the parent-specific monoallelic expression of genes in clusters. While imprinting perturbations are widely associated with developmental abnormalities, the intricate regional interplay between imprinted genes makes interpreting the contribution of gene dosage effects to phenotypes a challenging task. Using mouse models with distinct deletions in an intergenic region controlling imprinting across the Dlk1-Dio3 domain, we link changes in genetic and epigenetic states to allelic-expression and phenotypic outcome in vivo. This determined how hierarchical interactions between regulatory elements orchestrate robust parent-specific expression, with implications for non-imprinted gene regulation. Strikingly, flipping imprinting on the parental chromosomes by crossing genotypes of complete and partial intergenic element deletions rescues the lethality of each deletion on its own. Our work indicates that parental origin of an epigenetic state is irrelevant as long as appropriate balanced gene expression is established and maintained at imprinted loci.

Mammalian parental imprinting is a form of epigenetic regulation that causes genes to be expressed from only one chromosome homolog according to parent-of-origin[1,2]. Imprints entail the maintenance of germline-derived differential epigenetic marks, mostly in the form of DNA methylation, through to the somatic cells of the offspring. The resulting parent-specific signature serves as an imprinting control center (ICR) that regulates the monoallelic expression of multiple imprinted genes in a cluster[3,4]. Global as well as locus-specific alterations to ICRs, have emphasized that loss-of-imprinting results in reciprocal effects on imprinted genes with a biallelic expression of some genes within the cluster and biallelic repression at others[5–9]. Phenotypically, perturbations to individual genes were shown to exert effects in numerous developmental and physiological pathways[10,11]. Together, this has led to the prevailing notion in the field that imprinted genes

are dosage-sensitive. Yet, the intricate form of epigenetic control over the parent-specific expression of multiple genes in an imprinted cluster poses difficulties when trying to decipher the relative contribution of changes in imprinted gene dosage to the resulting physiological phenotypes.

One of the largest imprinted clusters in mammals is a 1.2 Mb domain encompassing the *Dlk1* and *Dio3* genes. Three protein-coding genes: *Dlk1*, *Rtl1*, and *Dio3* are exclusively expressed from the paternal allele, whereas multiple noncoding transcripts, including *Gtl2*, and its associated transcripts *Rian*, and *Mirg*, are expressed from the maternally inherited chromosome (Fig. 1a). A paternal-derived intergenic differentially methylated region (IG-DMR) was shown to play a key role in regulating parent-specific expression in this locus[12]. After implantation, a secondary DMR is established at the promoter of the *Gtl2*

[1]Department of Molecular Cell Biology, Weizmann Institute of Science, 7610001 Rehovot, Israel. [2]Department of Genetics, University of Cambridge, Cambridge CB2 3EH, United Kingdom. [3]These authors contributed equally: Ariella Weinberg-Shukron, Raz Ben-Yair. ✉e-mail: afsmith@gen.cam.ac.uk; yonatan.stelzer@weizmann.ac.il

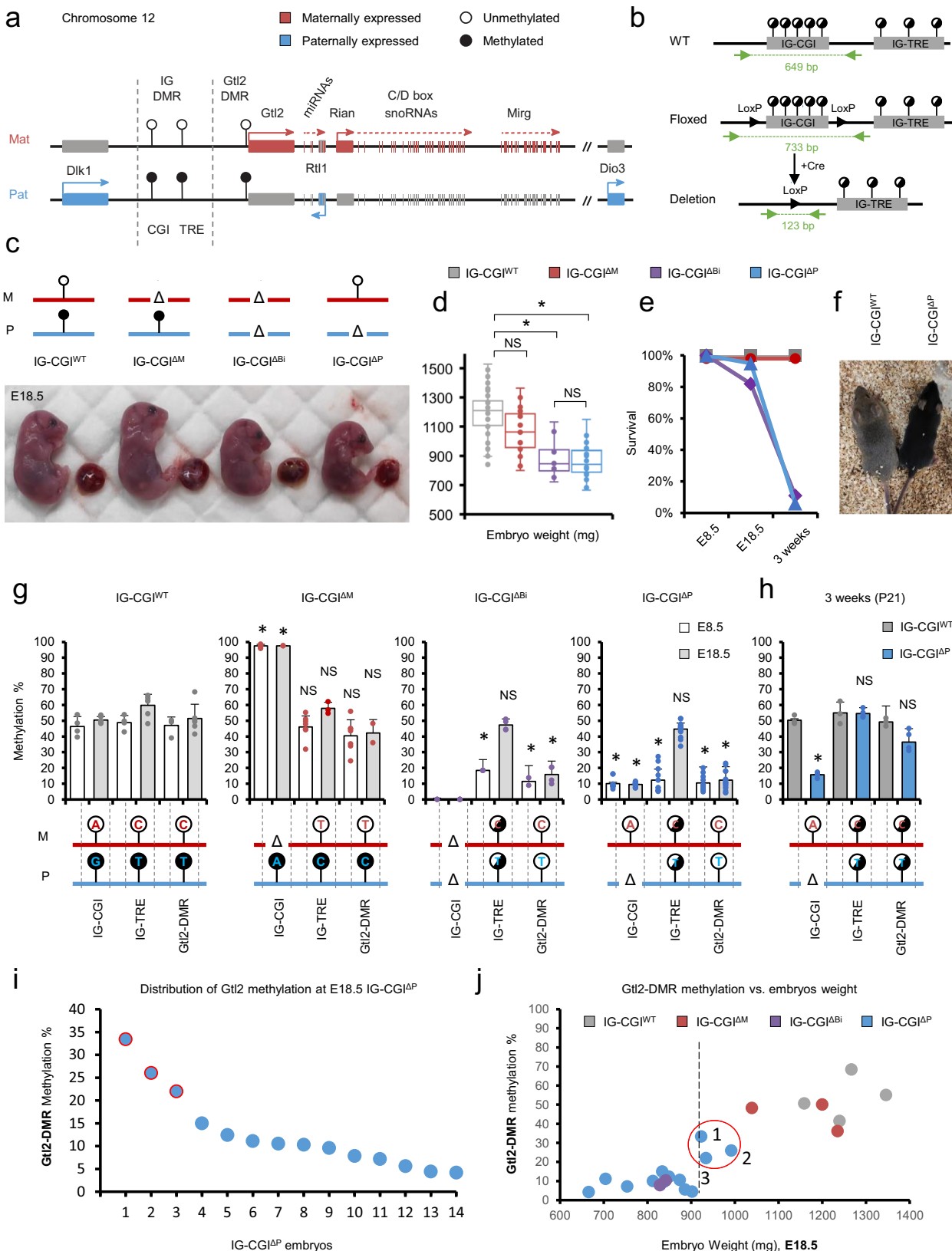

gene, sustaining its repression from the paternal allele. Maintenance of parent-specific epigenetic marking at the IG-DMR is crucial for normal development. Maternal deletion of the entire IG-DMR was shown to result in perinatal lethality, while paternal deletion was consistent with normal development[13,14]. Surprisingly, an isolated paternally derived deletion of a CpG island (CGI) located at the 5′ portion of the IG-DMR

was shown to result in the opposing paternal-to-maternal phenotype[15]. Insight into the underlying mechanism has emerged recently, as the IG-DMR was shown to comprise two distinct functional elements[16]. The upstream region of the IG-DMR contains a CGI which maintains a repressive chromatin landscape on the paternal allele[17]. Notably, this region included a Zfp57 binding site that is essential for maintaining

**Fig. 1 | Paternally transmitted deletion of the IG-CGI results in perinatal lethality. a** Schematic representation of the mouse Dlk1-Dio3 locus; Open lollipops–unmethylated region; black lollipops–methylated region. **b** Schematic representation of Cre-lox mediated targeting of the IG-CGI. Dashed green lines represent amplicons used for screening and genotyping. **c** Representative images of E18.5 embryos and their placentas for different genotypes. Corresponding genetic deletion and inferred methylation status of the IG-CGI are shown above. **d** E18.5 embryo weights per genotype. $N_{WT} = 75$, $N_{\Delta M} = 27$, $N_{\Delta Bi} = 14$, $N_{\Delta P} = 64$ biologically independent embryos; NS- not significant. Asterisks indicate statistical significance in comparison to WT using a one-way ANOVA (ΔBi: $p = 4.5e\text{-}12$, ΔP: $p = 9.4e\text{-}31$). Box plot minima = 948.7, 1047, 721.9, 665.9; maxima = 1528.5, 1366.4, 1132.3, 1149.7; center = 1217.5, 1184.2, 846.9, 843.15; for WT, ΔM, ΔBi, and ΔP respectively. Bounds of boxes show the 25th and 75th percentiles. Whiskers extend 1.5 times the interquartile range. **e** Survival plot per genotype. **f** Representative images of three weeks old viable IG-CGI$^{\Delta P}$ pup and wild-type (WT) littermate. **g, h** Methylation analysis by bisulfite pyrosequencing of three regulatory elements in embryos of all genotype groups. Inferred allele-specific methylation status based on SNPs (letters in lollipops) is depicted below. **g** $N_{WT} = 6,7$; $N_{\Delta M} = 9,7$; $N_{\Delta Bi} = 3,5$; $N_{\Delta P} = 11,17$ biologically independent embryos for E8.5 and E18.5, respectively. NS not significant. Asterisks indicate statistical significance in comparison to WT using a one-way ANOVA (IG-CGI$^{\Delta M}$: $p = 3.71e\text{-}11$ (E8.5), $p = 1.1e\text{-}7$ (E18.5) for IG-CGI. IG-CGI$^{\Delta Bi}$: $p = 9.2e\text{-}5$ (E8.5), $p = 0.001$ (E8.5), and $p = 0.0001$ (E18.5) for IG-TRE and Gtl2-DMR, respectively. IG-CGI$^{\Delta P}$: $p = 1.03e\text{-}9$ (E8.5) and $p = 3.48e\text{-}24$ (E18.5), $p = 1.34e\text{-}6$ (E8.5), $p = 1.39e\text{-}$ (E8.5), and $p = 6.7e\text{-}10$ (E18.5) for IG-CGI, IG-TRE, and Gtl2-DMR, respectively. **h** Number of surviving pups and controls $N = 3, 4$, biologically independent animals respectively. NS not significant. Asterisks indicate statistical significance in comparison to WT using a one-way ANOVA ($p = 4.58e\text{-}9$ for IG-CGI). Data are presented as mean values ± SD (**g, h**). **i** Dot-plot depicting IG-CGI$^{\Delta P}$ embryos ($x$ axis), arranged according to levels of methylation at the Gtl2-DMR ($y$ axis). **j** Weights of E18.5 embryos from different genotypes plotted against Gtl2-DMR methylation values. Circled in red are individual IG-CGI$^{\Delta P}$ embryos with >900 mg weight, corresponding to higher Gtl2-DMR methylation levels in **i**; $N_{WT} = 4$, $N_{\Delta M} = 3$, $N_{\Delta P} = 14$, and $N_{\Delta Bi} = 2$ biologically independent embryos. Pearson's correlation coefficient = 0.59523.

paternal methylation during the early stages of pre-implantation[5,6]. The remaining distal part of the IG-DMR was shown to bind pluripotency transcription factors in mouse embryonic stem cells (mESCs), exhibit active enhancer marks (H3K27ac), and nascent transcription[18,19]. It was therefore suggested to serve as a putative Transcriptional Regulatory Element (TRE), driving the expression of maternally inherited genes in the locus. Following implantation, this transcriptional activity prohibits the accumulation of de novo methylation, thus establishing the parent-specific Gtl2-DMR[20–22] (Fig. 1a). Together, the paradoxical effects imposed by distinct deletions within the IG-DMR represent an attractive experimental framework for dissecting the consequent impact of changes in gene dosage on embryonic phenotypes.

Here, we address the hierarchical interplay between the regulatory elements in this region by combining mouse models with complementary deletions, correlating allele-specific methylation, gene expression, and phenotypical outcome. The relationship between the regulatory hierarchy of these elements and the resulting epigenetic states on the two parental chromosomes has broader implications for our understanding of interactive regulatory modules in *cis*. Importantly, our data show that irrespective of the parental origin and epigenetic landscape of the IG-DMR, normal development strictly depends on maintaining a balanced expression between genes in the Dlk1-Dio3 locus, rather than parent-of-origin specific expression. Our work thereby provides a conceptual framework for understanding the emergence of epigenetic mechanisms controlling parent-specific gene expression.

## Results

### Paternal, but not maternal deletion of the IG-CGI results in perinatal lethality

We utilized CRISPR/Cas9 in mESCs to introduce LoxP sites flanking the CGI located at the 5′ portion of the IG-DMR (hereinafter termed IG-CGI$^f$), which were subsequently used to generate transgenic mice (Fig. 1b and Supplementary Fig. 1a, b). To assess the impact of parent-specific deletion of the IG-CGI on embryonic development, IG-CGI$^{f/f}$ mice were crossed with a transgenic strain expressing Cre recombinase under the germline-specific Vasa promoter. F1 mice from this cross are heterozygous for the deleted allele, but only in their germ cells. In turn, crossing F1 mice generated offspring with either paternal (IG-CGI$^{\Delta P}$), maternal (IG-CGI$^{\Delta M}$), or biallelic(IG-CGI$^{\Delta Bi}$) deletions of the IG-CGI (Supplementary Fig. 1c–e). Consistent with recent results[15], we observed noticeable size differences between the genotypes at embryonic day (E)18.5 (Fig. 1c and Supplementary Fig. 2a). While embryos with paternal or biallelic deletions were smaller and weighed less compared to wild-type (WT) counterparts, embryos with maternal deletion were indistinguishable from wild-type (Fig. 1d and

Supplementary Fig. 2b). Size differences were also evident in placentas, with embryos harboring paternal and biallelic deletions exhibiting significantly reduced placental weight (Supplementary Fig. 2c). To identify potential tissue level perturbations, we next performed a histological survey on embryonic and extra-embryonic tissues derived from E18.5 IG-CGI$^{\Delta P}$ and WT littermates. This analysis did not detect any gross cellular or morphological defects in IG-CGI$^{\Delta P}$ or IG-CGI$^{\Delta Bi}$ embryos relative to wild-type and IG-CGI$^{\Delta M}$ embryos, with the possible exception of dorsal brown fat, which appeared reduced in volume in IG-CGI$^{\Delta P}$ embryos (Supplementary Fig. 2d). Yet, we noted that this reduction in brown fat content was variable between embryos and tissue sections from different areas. Finally, while the overall structure of the placentas appeared intact in mutants compared to controls, we identified a marked reduction in absolute thickness of the labyrinthine layer (Supplementary Fig. 2e, f).

Monitoring postnatal survival identified that the vast majority of IG-CGI$^{\Delta P}$ and IG-CGI$^{\Delta Bi}$ neonates die within the first 24 hours. This was in contrast to IG-CGI$^{\Delta M}$ mice, which displayed normal long-term survival, were fertile, and lacked apparent developmental defects (Fig. 1e and Supplementary Fig. 2g). Surprisingly, we found that out of 17 litters and 62 WT littermates, 4 IG-CGI$^{\Delta P}$ and 2 IG-CGI$^{\Delta Bi}$ neonates survived and matured (Fig. 1f and Supplementary Fig. 2g). Together, our results confirmed the requirement of an intact paternal IG-CGI for normal mouse development. While our analysis could not identify major physiological defects in IG-CGI$^{\Delta P}$ or IG-CGI$^{\Delta Bi}$ embryos, both variations in brown fat content[23] and alterations in placenta structure and function could account for the perinatal lethality observed in these mutants.

### Paternal IG-CGI methylation is essential for the establishment of downstream DMRs

Parent-specific deletion of the IG-CGI offers a unique opportunity to study its role in establishing the regulatory landscape of this locus. To this end, we mated IG-CGI$^{f/f}$ with VASA-Cre mice to generate offspring with a deletion on either of the parental alleles in the germ cells in a C57BL/6 J background. These mice were subsequently reciprocally crossed with CAST/EiJ mice (Supplementary Fig. 1c–e). The resulting hybrid offspring allow the analysis of gene expression and DNA methylation at allelic resolution due to the high-density single nucleotide polymorphisms (SNPs) between these two strains. We used post-bisulfite pyrosequencing and PCR cloning analysis to assess methylation in post-implantation embryos. To control for developmentally associated changes in IG-DMR methylation, we analyzed bulk DNA and RNA from E8.5 whole-embryos or E18.5 tails, both shown to exhibit intact imprinting in WT[24]. At E8.5, mutant embryos appeared phenotypically indistinguishable from WT embryos, displaying similar size, clear headfolds, heart rudiment, and first somites (Supplementary

Fig. 3a). Bulk methylation measurements confirmed intermediate levels in the IG-CGI of WT embryos, corresponding to paternal, but not maternal, allelic methylation (Fig. 1g and Supplementary Fig. 3b). Methylation on the IG-TRE and Gtl2-DMR was likewise exclusively methylated on the paternal allele (Fig. 1g and Supplementary Fig. 3b). In IG-CGI$^{\Delta M}$ mutant embryos, the IG-CGI was fully methylated, reflecting the absence of the maternal copy, but downstream methylation at the IG-TRE and Gtl2-DMR remained unchanged and retained monoallelic methylation. In contrast, IG-TRE and Gtl2-DMR methylation levels decreased significantly in IG-CGI$^{\Delta P}$/IG-CGI$^{\Delta Bi}$ mutant embryos (Fig. 1g and Supplementary Fig. 3c, d). A similar overall trend was observed for the IG-CGI and Gtl2-DMR in E18.5 embryos (Fig. 1g and Supplementary Fig. 4a). However, a notable exception was the accumulation of variable methylation levels at the IG-TRE region on both parental alleles in E18.5 IG-CGI$^{\Delta P}$ embryos (Supplementary Fig. 4a). Together, our findings indicate that methylation at the IG-TRE and Gtl2-DMR on the paternal chromosome is dependent on the primary methylation marking of the paternal IG-CGI.

## Gtl2-DMR methylation levels correlate with IG-CGI$^{\Delta P}$ embryo weight and survival

We next analyzed methylation levels associated with the three regulatory elements in surviving IG-CGI$^{\Delta P}$ pups. Compared to both IG-CGI$^{\Delta P}$ and IG-CGI$^{\Delta Bi}$ E18.5 mutant embryos, the IG-CGI and IG-TRE maintained similar levels of methylation. But intriguingly, methylation at the secondary Gtl2-DMR increased substantially (Fig. 1h). We, therefore, hypothesized that methylation levels at the Gtl2-DMR may positively correlate with improved phenotype. Considering the low survival rates of IG-CGI$^{\Delta P}$ neonates (4/55), we asked whether variable methylation levels may already exist between prenatal embryos from a similar genotype and whether such changes may correlate with detectable phenotypical differences. Variation in Gtl2-DMR methylation levels was detected between individual E18.5 IG-CGI$^{\Delta P}$ embryos, with 3 out of 14 embryos exhibiting significantly increased methylation (>20%; Fig. 1i and Supplementary Fig. 4b). This was in contrast to IG-CGI and IG-TRE regions which overall appeared less variable (Supplementary Fig. 4b). Next, we analyzed WT, IG-CGI$^{\Delta M}$, IG-CGI$^{\Delta Bi}$, and IG-CGI$^{\Delta P}$ embryos at E18.5 for methylation on all three regulatory elements and recorded their weights. This analysis identified a correlation between Gtl2-DMR methylation levels and embryo weight, with the three embryos exhibiting >20% methylation also demonstrating increased weights (>900 mg; Fig. 1j). It is noteworthy that analyzing F1 hybrid IG-CGI$^{\Delta P}$ embryos at E18.5, identified that gain in Gtl2-DMR methylation occurred on both alleles independent of their parent-of-origin (Supplementary Fig. 4c).

Hence, taken together, while most embryos harboring a paternal deletion of the IG-CGI die around birth, we identified variable weights in prenatal embryos and a few instances of neonates that developed to adulthood. This phenotype was correlated with stochastic accumulation of methylation on both alleles of the Gtl2-DMR but not in IG-TRE DMR, which showed elevated methylation in all E18.5 embryos, irrespective of their weights and suggested a hierarchy of epigenetic events involving the three regulatory elements.

## Relationship between allele-specific expression and phenotypic variation in mutant embryos

Changes in DNA methylation at the three regulatory regions are predicted to affect the allelic expression of genes in the Dlk1-Dio3 locus. To systematically quantify this, we performed both bulk and allele-specific expression analysis on E8.5 F1 hybrid embryos harboring parent-specific deletions of the IG-CGI. Expression of *Dlk1* was significantly reduced in IG-CGI$^{\Delta P}$ and IG-CGI$^{\Delta Bi}$ embryos compared to WT or IG-CGI$^{\Delta M}$ in which it is predominantly expressed from the paternal allele (Fig. 2a). Reciprocally, maternal genes (*Gtl2*, *Rian*, and *Mirg*) were upregulated in IG-CGI$^{\Delta P}$ and IG-CGI$^{\Delta Bi}$ embryos compared to controls,

as a result of switching to biallelic expression (Fig. 2a). Nearly identical results were found when comparing E18.5 embryos from the four different genotypes (Fig. 2b). Given the confirmed relationships between genotype, epigenetic state of regulatory elements, and gene expression effects, we next addressed whether the expression ratio between genes in the locus could predict phenotypic outcome as measured by embryonic weight in E18.5. We collected embryos from the four different genotypes and plotted the expression values of *Dlk1* and *Gtl2* per individual embryo. Overall, this analysis robustly distinguished IG-CGI$^{\Delta P}$ or IG-CGI$^{\Delta Bi}$ embryos and control WT or IG-CGI$^{\Delta M}$ embryos (Fig. 2c). Moreover, it also uncovered a subset of IG-CGI$^{\Delta P}$ embryos (3/15) that exhibited higher *Dlk1* to *Gtl2* expression ratios corresponding to increased weight values (>900 mg; Fig. 2d and Supplementary Fig. 4d). Finally, a tight correlation in the expression patterns of *Gtl2* and the associated transcripts, *Rian*, and *Mirg* was observed across the 4 different genotypes, strengthening the established notion that these maternally expressed transcripts are co-regulated (Fig. 2e, f). Together, our analysis identified correlations between changes in gene dosage and developmental phenotypes. Whereas biallelic expression of *Gtl2* and repression of *Dlk1* are associated with reduced weight and early postnatal lethality, de novo compensatory methylation at the Gtl2-DMR is shown to result in an increased *Dlk1* to *Gtl2* expression ratio, increased body weight, and improved outcome.

## Synthesis of gene dosage effects between two distinct IG-DMR mouse deletion models

In light of the findings described above and the enhanced understanding of the parent-specific regulatory landscape, we attempted to solve a puzzling and apparently contradictory characteristic of the two different deletion models: in the current study, maternal deletion of the IG-CGI is inconsequential for both gene expression, development and survival, while paternal deletion causes dysregulation in gene expression and is typically lethal. However, deletion of the entire IG-DMR[13,14], which includes both the IG-CGI and IG-TRE, is characterized by similar deleterious effects when the maternal copy is deleted, while the paternal deletion is inconsequential (Fig. 3). Synthesizing the result of the two genetic models shows that normal development cannot occur with a biallelic expression of maternal genes and repression of *Dlk1* or with a biallelic expression of *Dlk1* and repression of maternal transcripts (Fig. 3). Independent of the genetic and epigenetic state of the IG-DMR, in both models as in WT, concurrent monoallelic expression of *Dlk1* and maternal transcripts is consistent with normal development. This raises the question of whether it is an expression from the appropriate parental chromosome or a balanced expression between genes per se that is required for postnatal survival. We, therefore, set out to unequivocally distinguish between these two scenarios using a genetic model incorporating the IG-DMR deletion.

## Generation of viable mice carrying inverted allelic gene expression

Crossing IG-CGI$^{\Delta/WT}$ males with IG-DMR$^{\Delta/WT}$ females are predicted to generate a scenario of double deletion in a quarter of the embryos leading to flipped imprinting regulation, with maternal genes expressed from the paternal allele and vice versa (See Fig. 4a and breeding strategy in Supplementary Fig. 5a). We hypothesized that inheritance of both altered alleles, each on its own associated with postnatal lethality, would lead to viable pups if balanced gene dosage was paramount over parent-of-origin regulation and expression. To this end, we crossed IG-CGI$^{\Delta/WT}$ males with IG-DMR$^{\Delta/WT}$ females and obtained 18 pups, of which seven were genotyped as IG-DMR$^{\Delta M}$; IG-CGI$^{\Delta P}$ doubles deletions. Remarkably, these pups developed normally, were viable, and did not exhibit any overt deficiencies. This, is in contrast to isolated single deletion IG-DMR$^{\Delta M}$ and IG-CGI$^{\Delta P}$ littermates that did not survive postnatally (Fig. 4b). Notably, IG-DMR$^{\Delta M}$; IG-CGI$^{\Delta P}$ double deletion males were ~20% lighter on average than WT males by

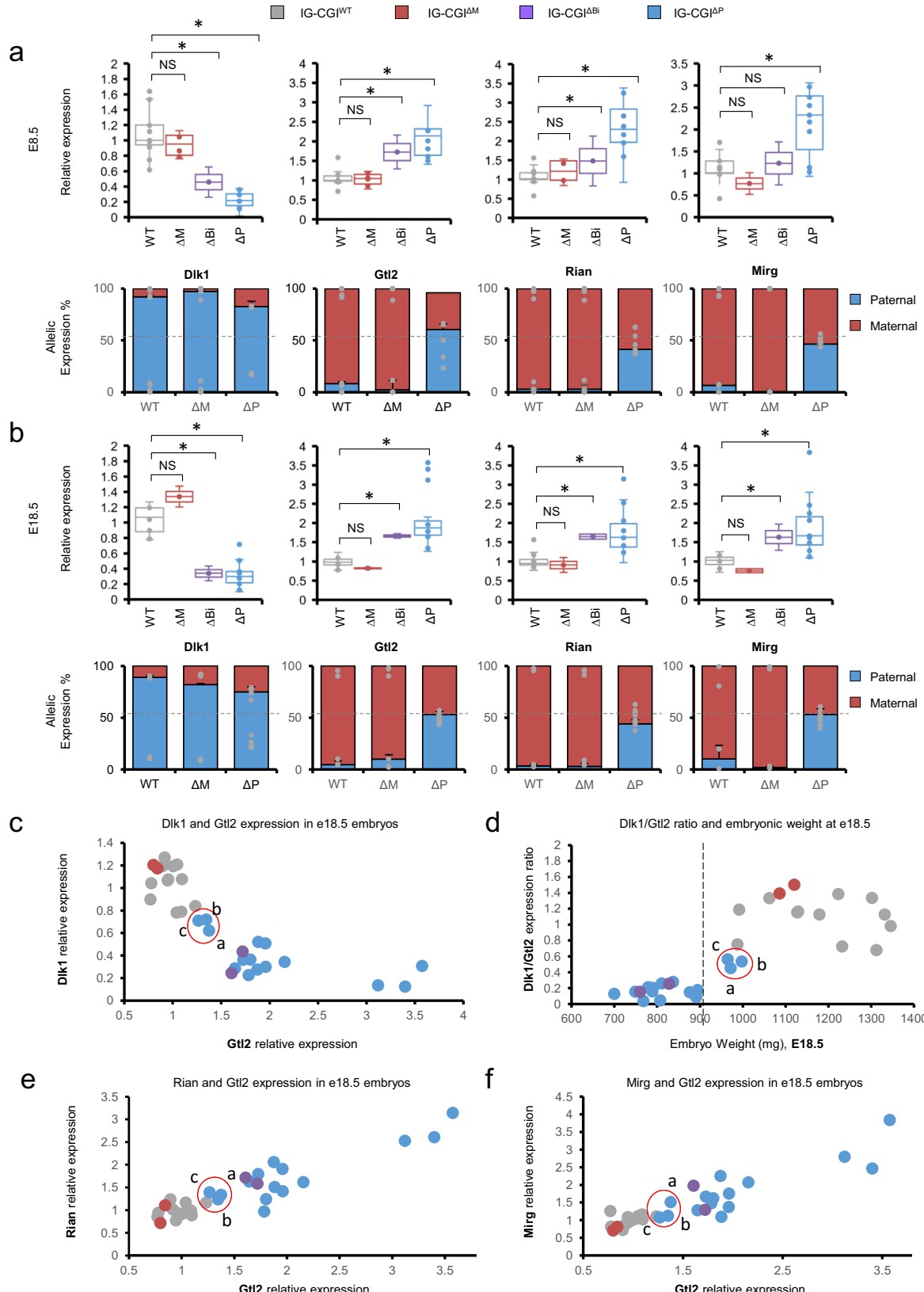

four weeks, while females were overall indistinguishable in weight (Supplementary Fig. 5c, d). The reciprocal cross of IG-CGI$^{Δ/WT}$ females with IG-DMR$^{Δ/WT}$ males resulted in viable offspring from all genotypes including double deletion pups. This is expected since mice harboring single deletions are viable (Supplementary Fig. 5b. $N = 8, 10, 6$ for IG-DMR$^{ΔP}$, IG-CGI$^{ΔM}$ and IG-DMR$^{ΔP}$; IG-CGI$^{ΔM}$ respectively).

To understand the regulatory configuration that allows these animals to develop normally, we carried out pyrosequencing-mediated DNA methylation analysis at different time points in development. Our analysis confirms that the maternal IG-DMR deletion encompassing both the IG-CGI and IG-TRE leads to a methylated Gtl2-DMR, in contrast to the maternal deletion of the IG-CGI alone we initially described

**Fig. 2 | Gene expression effects following the parent-specific deletion of IG-CGI.** **a** Box-plots represent quantitative real-time PCR (qRT-PCR) analysis of representative genes in the Dlk1-Dio3 region in E8.5 embryos from different genotypes. Shown is mean relative fold change ± s.d; $N_{WT}$ = 20, $N_{ΔM}$ = 6, $N_{ΔBi}$ = 3, $N_{ΔP}$ = 12 biologically independent embryos. Column graphs represent a relative allelic expression of the Dlk1-Dio3 genes, as measured by pyro SNP analysis ($N_{WT}$ = 6, $N_{ΔM}$ = 5, $N_{ΔP}$ = 5 biologically independent embryos). Dlk1 Box plot minima = 0.6, 0.76, 0.26, 0.02; maxima = 1.64, 1.13, 0.66, 0.37; center = 1, 0.95, 0.46, 0.22. Gtl2 Box plot minima = 0.72, 0.78, 1.29, 1.41; maxima = 1.58, 1.23, 2.16, 2.92; center = 1, 1.05, 1.73, 2.14. Rian Box plot minima = 0.57, 0.84, 0.83, 0.93; maxima = 1.56, 1.53, 2.13, 3.38; center = 1.02, 1.22, 1.48, 2.31. Mirg Box plot minima = 0.43, 0.52, 0.74, 0.93; maxima = 1.71, 1.01, 1.72, 3.06; center = 1.02, 0.77, 1.23, 2.33; for WT, ΔM, ΔBi, and ΔP, respectively. Bounds of boxes show the 25th and 75th percentiles. Whiskers extend 1.5 times the interquartile range. NS not significant. Asterisks indicate statistical significance in comparison to WT using a one-way ANOVA (Dlk1 ΔBi: $p$ = 0.002; ΔP: $p$ = 7.5e-11. Gtl2 ΔBi: $p$ = 2.8e-5; ΔP: $p$ = 5.2e-10. Rian ΔBi: $p$ = 0.03; ΔP: $p$ = 2.6e-8. Mirg ΔP: $p$ = 0.0002). **b** Identical analysis as in **a**, performed in E18.5 embryos; $N_{WT}$ = 12,

$N_{ΔM}$ = 3, $N_{ΔBi}$ = 3, $N_{ΔP}$ = 15 biologically independent embryos in qRT-PCR box-plots; Dlk1 Box plot minima = 0.78, 1.2, 0.24, 0.1; maxima = 1.27, 1.47, 0.44, 0.72; center = 1.07, 1.34, 0.34, 0.3. Gtl2 Box plot minima = 0.77, 0.8, 1.61, 1.27; maxima = 1.23, 0.84, 1.72, 3.57; center = 0.98, 0.82, 1.66, 1.88. Rian Box plot minima = 0.77, 0.72, 1.58, 0.97; maxima = 1.56, 1.1, 1.71, 3.14; center = 0.95, 0.91, 1.65, 1.63. Mirg Box plot minima = 0.72, 0.71, 1.29, 1.09; maxima = 1.25, 0.81, 1.97, 3.84; center = 1.02, 0.76, 1.63, 1.67; for WT, ΔM, ΔBi and ΔP, respectively. Bounds of boxes show the 25th and 75th percentiles. Whiskers extend 1.5 times the interquartile range. NS not significant. Asterisks indicate statistical significance in comparison to WT using a one-way ANOVA (Dlk1 ΔBi: $p$ = 1.57e-5; ΔP: $p$ = 4.4e-11. Gtl2 ΔBi: $p$ = 1.16e-6; ΔP: $p$ = 3.13e-5. Rian ΔBi: $p$ = 0.0002; ΔP: $p$ = 0.0002. Mirg ΔBi: $p$ = 0.0002; ΔP: $p$ = 0.0003). $N$ = 5 biologically independent embryos per genotype group for allelic expression column graphs. **c–f** qRT-PCR based relative expression values of genes in the Dlk1-Dio3 region normalized to WT, measured in E18.5 embryos. Circled in red are IG-CGI$^{ΔP}$ embryos with >900 mg weight. Dot colors indicate genotype as in **a**, **b**. Pearson's correlation coefficient = −0.6807 (**c**), 0.7477 (**d**), 0.8860 (**e**), 0.8858 (**f**).

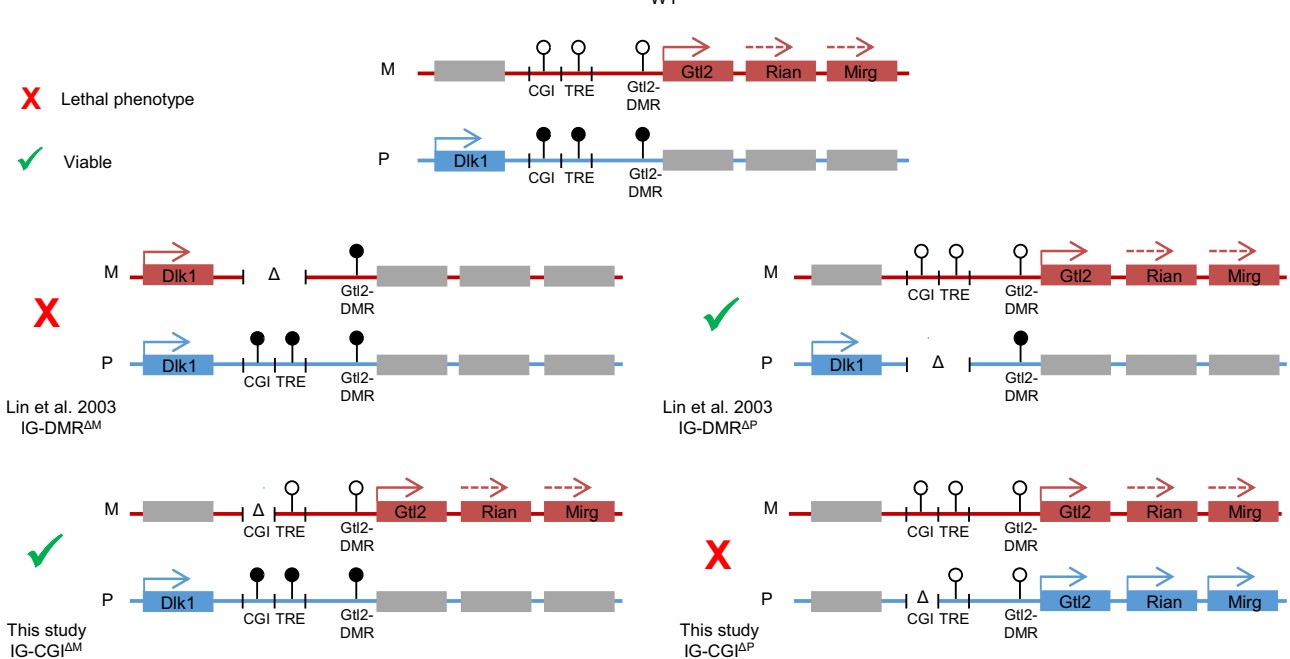

**Fig. 3 | Comparing epigenetic and allelic effects between two IG-DMR mouse deletion models.** Colored boxes represent expressions from maternal (red) and paternal (blue) alleles. Gray boxes represent allelically-repressed genes. Lollipops represent methylated (black) and unmethylated (white) regulatory elements.

(Fig. 4c, d and Fig. 1g and Supplementary Figs. 3c, 4a). This result is compatible with the IG-TRE element positively regulating maternal gene expression. When the IG-TRE element is absent, *Gtl2* is not expressed, and the region acquires de novo methylation following implantation, similar to other non-transcribed genomic regions. In IG-DMR$^{ΔM}$; IG-CGI$^{ΔP}$ embryos and postnatal animals, we found intermediate methylation levels at the Gtl2-DMR. Together, these results reflected the combination of maternal-to-paternal epigenotype switching on the maternal chromosome, and paternal-to-maternal epigenotype switching on the paternally inherited chromosome. (Fig. 4c, d, and Supplementary Fig. 5e).

Consistent with results from IG-CGI$^{ΔP}$ (Fig. 1g) we found that also in IG-DMR$^{ΔM}$;IG-CGI$^{ΔP}$ embryos and their IG-CGI$^{ΔP}$ littermates both IG-TRE alleles were unmethylated early in development, and de novo methylation accumulated on some paternal alleles between E8.5 and E18.5. Importantly, this did not appear to affect Gtl2-DMR methylation in embryos and postnatal animals (Fig. 4c, d, and Supplementary Fig. 5e). Gene expression analysis confirmed the reciprocal gene dosage effects between IG-CGI$^{ΔP}$ and IG-DMR$^{ΔM}$, with the former

exhibiting lower levels of *Dlk1* relative to WT littermates (~5 folds) together with elevated levels of all maternal transcripts at this locus (~1.5 folds), while the latter showing the reciprocal effects (Fig. 4e). Importantly, in agreement with this observed regulatory switch, we found that both paternal and maternal gene expression is restored to normal levels in IG-DMR$^{ΔM}$; IG-CGI$^{ΔP}$ double deletion embryos (Fig. 4e and Supplementary Fig. 5f). Our results, therefore highlight a fundamental requirement for balanced gene dosage in the Dlk1-Dio3 locus for proper embryonic growth and survival regardless of which parental allele genes are expressed from.

## Discussion

Regulation of imprinted gene expression is established via gamete-specific epigenetic marking that initiates distinct regulatory hubs on each parental chromosome. This differential signal influences *cis* interactions and binding of *trans*-acting factors that, in turn, determine the allelic expression of both nearby and remote genes. In this study, we dissected the sequential activity of epigenetic mechanisms that bring about parent-specific gene expression in the Dlk1-Dio3 imprinted

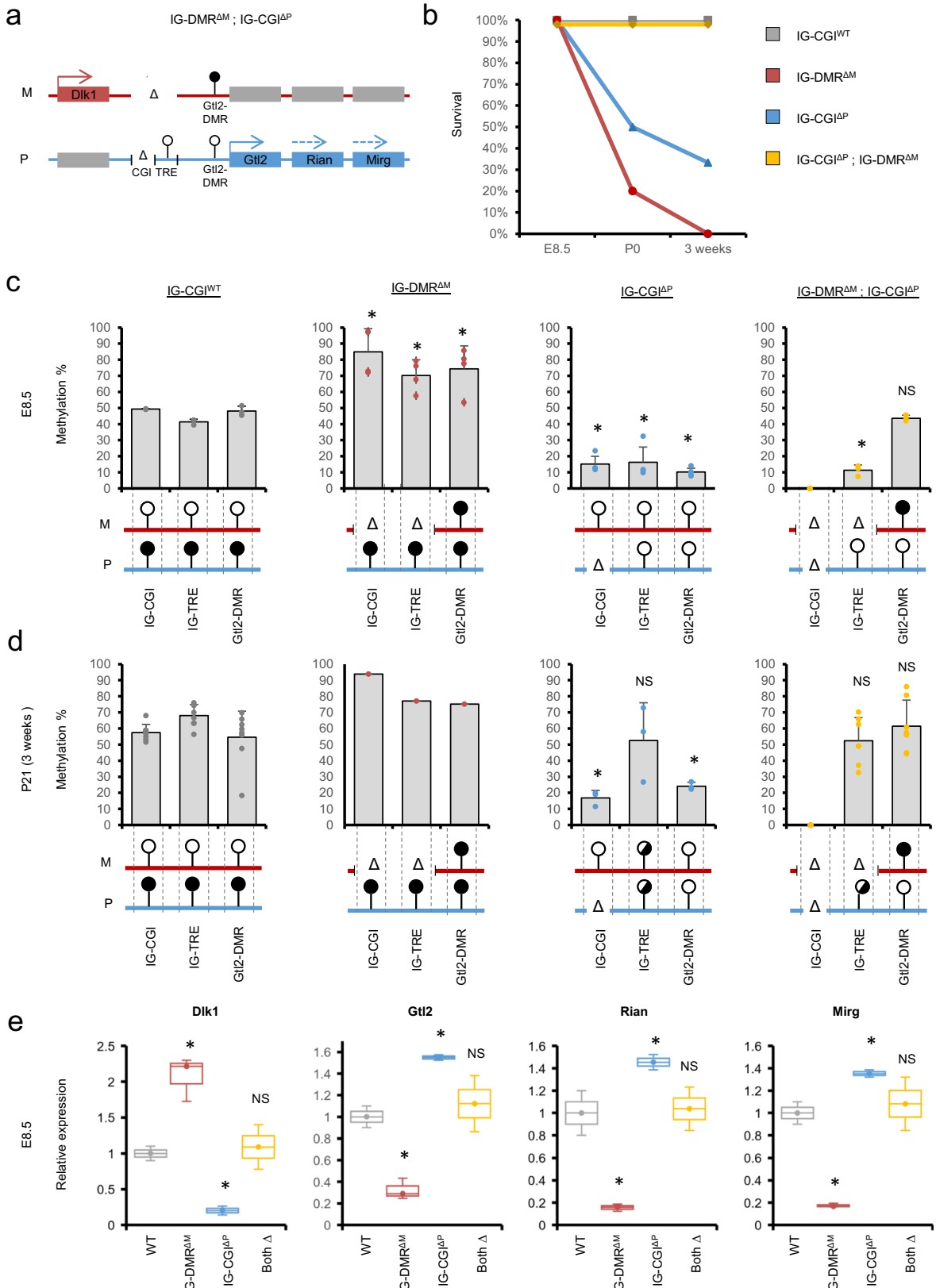

cluster during embryonic development. We devised an experimental framework including mouse models with distinct genetic alteration in the regulatory IG-DMR and tightly controlled allelic readout from single embryos. This allowed robustly linking of epigenetically controlled changes in imprinted gene dosage to associated developmental phenotypes.

Our data suggest that methylation of the paternal IG-CGI allele provides an initial and essential signal for methylation of the IG-TRE and *Gtl2* promoter in *cis* (Fig. 5). This is supported by the unmethylated status of the IG-TRE and Gtl2-DMR in E8.5 IG-CGI^ΔP embryos. Synthesizing results of previous work[19], we propose an in vivo model in which methylation of the IG-CGI triggers the transcriptional repression of a

**Fig. 4 | Generation of a mouse model carrying balanced, inverted regulation of allele-specific expression. a** Schematic representation of the predicted outcome of crossing IG-CGI$^{\Delta P}$ and IG-DMR$^{\Delta M}$ mouse strains. **b** Survival plots of different genotype groups. **c**, **d** Methylation analysis of the three regulatory elements using bisulfite pyrosequencing in E8.5 embryos (**c**) and P21 (3 weeks) postnatal pups (**d**). N(WT) = 3,8, N(IG-DMR$^{\Delta M}$) = 4,1, N(IG-CGI$^{\Delta P}$) = 4,3, N(IG-DMR$^{\Delta M}$; IG-CGI$^{\Delta P}$) = 3,7 biologically independent embryos for E8.5 and P21 postnatal pups, respectively. Data are presented as mean values ± SD. NS not significant. Asterisks indicate statistical significance in comparison to WT using a one-way ANOVA (IG-DMR$^{\Delta M}$: $p = 0.002$, $p = 0.001$, $p = 0.01$ for IG-CGI, IG-TRE, and Gtl2-DMR, respectively. IG-CGI$^{\Delta P}$: $p = 1.72e\text{-}5$ (E8.5) and $p = 6.66e\text{-}7$ (P21), $p = 0.003$ (E8.5), $p = 9.15e\text{-}7$ (E8.5), $p = 0.01$ (P21) for IG-CGI, IG-TRE and Gtl2-DMR respectively. IG-DMR$^{\Delta M}$; IG-CGI$^{\Delta P}$: $p = 1.15e\text{-}6$ for IG-TRE at E8.5). **e** qRT-PCR analysis of representative genes in the Dlk1-Dio3

region in E8.5 embryos of different genotype groups. $N = 3$ biologically independent embryos per group, colored as in **b**. Dlk1 Box plot minima = 0.9, 1.72, 0.14, 0.78; maxima = 1.1, 2.3, 0.26, 1.4; center = 1, 2.21, 0.2, 1.09. Gtl2 Box plot minima = 0.9, 0.25, 1.52, 0.86; maxima = 1.1, 0.43, 1.58, 1.38; center = 1, 0.29, 1.55, 1.12. Rian Box plot minima = 0.8, 0.12, 1.39, 0.84; maxima = 1.2, 0.19, 1.52, 1.23; center = 1, 0.16, 1.45, 1.04. Mirg Box plot minima = 0.9, 0.16, 1.32, 0.84; maxima = 1.1, 0.19, 1.39, 1.32; center = 1. 0.17, 1.35, 1.08; for WT, IG-DMR$^{\Delta M}$, IG-CGI$^{\Delta P}$ and IG-DMR$^{\Delta M}$;IG-CGI$^{\Delta P}$ respectively. Bounds of boxes show the 25th and 75th percentiles. Whiskers extend 1.5 times the interquartile range. NS not significant. Asterisks indicate statistical significance in comparison to WT using a one-way ANOVA (Dlk1 $p = 0.004$, 0.0003; Gtl2 $p = 0.001$, 0.0007; Rian $p = 0.001$, 0.02; Mirg $p = 0.0001$, 0.004; for IG-DMR$^{\Delta M}$ and IG-CGI$^{\Delta P}$ respectively).

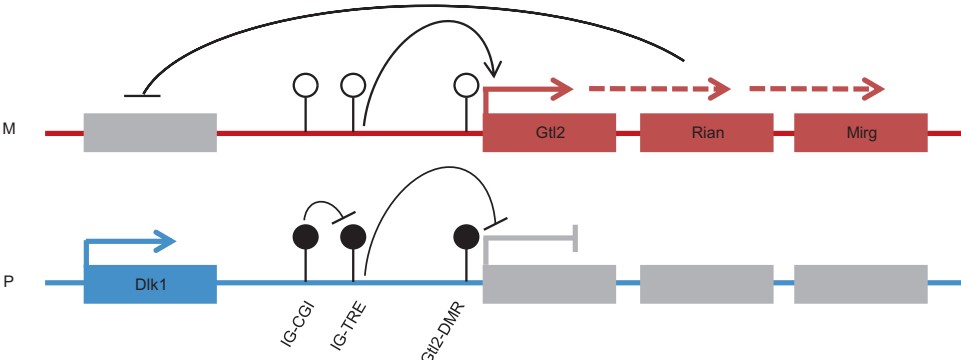

**Fig. 5 | An integrated model depicting allele-specific *cis*-regulation at the Dlk1-Dio3 region.** Paternal allele: the IG-CGI serves as the primary signal to repress the IG-TRE in *cis* via DNA methylation, subsequently preventing transcription of the downstream gene *Gtl2*. Following implantation, lack of *Gtl2* transcription results in de novo methylation of the Gtl2-DMR ensuring its repression and allowing *Dlk1*

expression from the same chromosome. Maternal allele: unmethylated, the IG-TRE element promotes the downstream transcription of the *Gtl2* polycistronic transcript, which in turn prevents accumulation of de novo methylation on the Gtl2-DMR. Our data are most consistent with the *Gtl2* polycistronic transcript repressing Dlk1 in cis via a yet uncharacterized mechanism.

downstream enhancer element (IG-TRE), driving the expression of the *Gtl2* polycistronic transcript. During the pre-implantation period, and similar to other genomic regions that lack transcriptional activity, de novo methylation of the *Gtl2* promoter serves to permanently repress its expression from the paternal allele (Fig. 5). This model fully explains what might appear as contradictory findings by Lin et al.[13,14] and the mouse model presented here and in a recent study[15]. In the current model, when the repressive IG-CGI element was removed from the paternal allele, the intact IG-TRE element facilitated erroneous transcription of Gtl2. But in the scenario of complete deletion of the IG-DMR, the lack of IG-TRE element precluded transcription of *Gtl2* from the paternal allele, mimicking WT situation of monoallelic expression of *Gtl2* and *Dlk1* (Fig. 3, compare right panels).

While genetic depletion of the IG-TRE is sufficient to induce transcriptional silencing and methylation of the *Gtl2* promoter, it remains to be determined whether methylation of the IG-TRE plays any role in modulating *Gtl2* expression. Indeed, sporadic accumulation of methylation at the IG-TRE did not affect methylation and expression of *Gtl2* in E18.5 and postnatal animals (See Figs. 1g, 4d, and Supplementary Fig. 5e). In contrast, sporadic promoter methylation was sufficient to reduce Gtl2 transcription. Therefore, it is possible that enhancer activity of IG-TRE is restricted to pre-implantation stages, after which, *Gtl2* expression is solely dependent on the epigenetic state of its promoter. In such a model, methylation of the IG-TRE and Gtl2-DMR merely reflects their transcriptional activity in different stages of development. This is in contrast to a recently proposed instructive regulation model imposed, in embryonic stem cells, by the antagonistic activity of Tet and Dnmt enzymes[16,25].

Beyond the sequential regulatory hierarchy described above, our data strongly support the existence of a crosstalk between maternal RNAs and Dlk1 expression *in cis*. As in the WT situation, all genetic

models used in this study showed exclusive expression of either the polycistronic RNA (including *Gtl2*, *Rian*, and *Mirg*) or *Dlk1*, never both, on the same chromosome. Importantly, this was irrespective of the genetic or epigenetic manipulation imposed on the IG-DMR. Deleting both IG-CGI and IG-TRE elements leads to repression of *Gtl2* and biallelic expression of *Dlk1*. Conversely, deletion of only the IG-CGI results in biallelic expression of *Gtl2* and repression of *Dlk1*. Considering the implication of long noncoding RNAs in gene silencing[22,26,27], our model is most consistent with maternal RNAs repressing Dlk1 in *cis* (Fig. 5). This notion is further supported by previous work demonstrating that *Dlk1* knockout does not affect the dosage of other genes in the locus[23,28,29].

Parent-specific perturbation of genes in the Dlk1-Dio3 imprinted locus was shown to exert a wide array of developmental and growth defects[12]. The intricate *cis* and *trans* interactions between genes in the locus, including those involving imprinted miRNAs[30–33] further complicate the interpretation of phenotypes associated with genetic or epigenetic IG-DMR perturbations. For example, genetic models presented in this study do not easily predict previously shown developmental phenotypes associated with the individual knockout of either *Dlk1* or *Gtl2*[34–42]. The latter exhibits a complex parent-of-origin-dependent phenotype, with other genes in the locus variably affected depending on inheritance mode, developmental stage, and tissue analyzed[34]. Yet synthesizing results from various IG-DMR genetic manipulations presented in this study implies that rather than combined effects of individual gene perturbations, it is the balanced expression between genes and their exquisite dosage control that is crucial for normal development[23].

Deviation from balanced gene expression, either by double-dosage of maternal RNAs and repression of *Dlk1*, or vice versa, leads to perinatal lethality. Phenotypically, it remains to be addressed whether

the placenta, embryo, or combination of both contribute to the dele-
terious developmental effects. Indeed, damaged fetal capillaries, as
reported in *Rtl1* KO placentas, could result in abnormal placenta
function and cause fetal lethality[43–45]. However, our results show that
compensatory methylation on *Gtl2* that partially increases the *Dlk1* to
maternal gene expression ratio can further restore viability in neonates
and even allow development and growth to proceed normally. More
strikingly, we demonstrate that experimentally flipping the parental
origin of the expression while retaining the balanced expression of
genes in the locus results in synthetic rescue of the perinatal lethality
presented when the same genes are expressed in a non-balanced
manner. The latter results indicate that the correct parent-of-origin
imprinting pattern is secondary to balanced gene dosage at this large
domain containing multiple reciprocally imprinted genes. Notably, the
described experimental design does not allow to distinguish whether
the resulting phenotypes are caused by varying ratios between
maternal to paternal genes or changes in absolute gene expression
since both are most likely affected by the manipulation.

Many imprinted genes exhibit tissue-specific expression. For
example, we have recently documented temporal and cell-state
dependent expression dynamics of imprinted genes during mouse
gastrulation[46]. In such context, when differential expression between
cell types can sometimes reach a hundred folds, it is not clear how
relatively mild effects of switching from monoallelic to biallelic
expression can introduce significant effects. On the other hand, as
exemplified for the Dlk1-Dio3 region, antagonistic effects entail the
robust repression of some genes in the locus, potentially resulting in
more severe effects. Intriguingly, nearly all imprinted clusters identi-
fied to date were shown to contain such reciprocal parent-specific
effects, often involving noncoding RNAs and complex *cis* and *trans*
interaction between genes in the region. In this respect, DNA methy-
lation appears to serve as a primary and robust maintenance
mechanism that ensures balanced expression between genes[47]. The
notion that parent-of-origin is irrelevant as long as balanced gene
expression is maintained in the specific developmental context has
recently been shown for another imprinted locus in vivo. An intercross
model of *Zdbf2* loss and gain of function similarly demonstrated that
the developmental phenotype is dose-dependent and irrelevant to
parent-of-origin[48]. On the other hand, allele switching at the *Peg3*
domain resulted in overall similar phenotypes to the WT, but did show
some differences in gene expression, suggesting potential non-
redundant roles contributed by the maternal and paternal
chromosomes[49]. It would be interesting to explore whether conserved
imprinted loci retain balanced gene dosage at the single cell level via
alternative mechanisms, in animals lacking epigenetic imprinting.

## Methods

### mESCs cell culture

V6.5 mouse embryonic stem cells (mESCs, Jaenisch lab, MIT.
RRID:CVCL_C865) were cultured at 37 °C with 5% $CO_2$, on plates coated
with 0.2% gelatine on irradiated mouse embryonic fibroblasts (MEFs,
DR4), in standard ESCs medium: (500 ml) DMEM (Gibco cat#41965-
039) supplemented with 20% US certified FBS (Biological Industries
cat#014-001-1 A), 10 μg recombinant leukemia inhibitory factor (LIF),
0.1 mM beta-mercaptoethanol (Gibco cat#31350-010), penicillin/
streptomycin 1 mM (Biological Industries cat#03-031-1B), L-glutamine
(Biological Industries cat#03-020-1B) and 1% nonessential amino acids
(Biological Industries cat#01-340-1B). For chimeras mESCs were cul-
tured on gelatin-coated plates with ESCs medium supplemented with
1 μM PD0325901 (Sigma-Aldrich cat#PZ0162) and 3uM CHIR99021
(Sigma-Aldrich cat#SML1046).

### Generation of IG-CGI floxed mESCs

To establish mESCs harboring loxP sites flanking the IG-CGI (IG-CGI[f/f]),
targeting vectors and CRISPR/Cas9 plasmids were co-transfected into
mESCs using Xfect mESC Transfection Reagent (Clontech Laboratories
cat#631320), according to the provider's protocol. sgRNAs and 5′ and
3′ homology arm sequences were cloned into *px330-BFP* and *px330-
GFP* vectors under U6 promoter (Addgene plasmid #98750, Wu 2013).
48 hours following transfection, cells were FACS sorted for double
positive BFP and GFP expression and plated on MEF feeder plates.
Single colonies were further analyzed for proper and single integration
and for male sex chromosomes by PCR analysis and Sanger sequen-
cing. The sgRNAs, homology arms, and genotyping primer sequences
for generating the IG-CGI[f/f] allele are listed in Supplementary
Tables 1 and 2.

### ES-Blastocyst injections and generation of IG-CGI floxed repor-
ter mice

Blastocyst injections were performed in BDF2 diploid blastocysts,
harvested from hormone-primed BDF1 4-week-old females. In brief,
4–5 week-old B6D2F1 females were hormone primed by an intraper-
itoneal injection of pregnant mare serum gonadotropin (PMSG, Pro-
Spec cat#HOR-272) followed 46 hours later by an injection of human
chorionic gonadotropin (hCG, Sigma-Aldrich cat#C1063-10VL).
Embryos were harvested at the zygote stage and cultured in a CO2
incubator until the blastocyst stage. Approximately ten cells were
injected into the blastocoel cavity of each embryo using a Piezo
micromanipulator (Prime Tech cat#PMM4G). Approximately 20 blas-
tocysts were subsequently transferred to each recipient female; the
day of injection was considered as 2.5 days postcoitum (DPC). Mice
were handled in accordance with institutional guidelines and approved
by the Institutional Animal Care and Use Committee (IACUC).

For germline transmission, male chimera mice were mated to
C57BL/6 females and the ones that gave birth to agouti pups (F1) had
the germline transmitted floxed IG-CGI allele. Male and female off-
spring carrying the IG-CGI[f/f] allele were bred and crossed with each
other until a homozygote IG-CGI[f/f] line was established (F2). The
homozygote IG-CGI[f/f] line was then crossed to the homozygote Ai14
Rosa26-lsl-tdTomato line (Jackson Laboratory stock#007914) until a
double homozygote IG-CGI[f/f];Rosa26-lsl-tdTom line was established.

### Mice lines

Vasa-Cre mice (FVB-Tg(Ddx4-cre)1Dcas/J) were obtained from
the Jackson Laboratory (stock#006954). CAST/EiJ (RRI-
D:IMSR_JAX:000928) mice were obtained from MRC Harwell Institute,
and maintained under a 12 hr light–dark cycle at 22 °C degrees (±2 °C)
and 55% humidity (±10%). Mice were monitored for health and activity
and were given ad libitum access to water and standard mouse chow.
For pure and hybrid breeding experiments, mice were mated at
8–12 weeks of age. F2 embryos harboring the deletion allele were
analyzed at different ages. All animal experiments were performed
according to the Animal Protection Guidelines of Weizmann Institute
of Science, Rehovot, Israel, and in accordance with the Animals (Sci-
entific Procedures) Act 1986 Amendment Regulations 2012 following
ethical review by the University of Cambridge Animal Welfare and
Ethical Review Body. Animal experiments were approved by relevant
Weizmann Institute IACUC (#39401117-3 and #00080118-2) and UK
Home Office project license #PC213320E. All efforts were made to
minimize animal discomfort.

### Genotyping

For genotyping, DNA was extracted from the tail tip using a solution
containing NaOH 1 M and EDTA 0.5 M pH8.0 in DDW, incubated for
1 hour at 95 °C and neutralized in Tris-HCl 40Mm pH5.0. Alternatively,
DNA was isolated by lysing cells in lysis buffer (0.1 M Tris buffer, 0.2 M
NaCl, 0.005 M EDTA, 0.2% SDS) with 10 mg/ml Proteinease K at 55 °C,
precipitated with Iso-Propanol, washed with 70% Ethanol and resus-
pended in TE buffer. Genotyping of mouse strains and alleles was done
by PCR, primers are listed in Supplementary Table 2.

## Embryo analysis

Embryos harboring the IG-CGI deletion allele were analyzed at different ages. At embryonic day E18.5 measurements of embryo weight (mg), length (cm), and placenta weight (mg) were recorded. At E18.5 bulk DNA and RNA from the tail were analyzed for methylation and gene expression profiles, representing the canonical imprinting pattern at the Dlk1-Dio3 region (as represented in Fig. 1a). At E8.5 the whole embryo was used for bulk DNA and RNA purification and analysis. Postnatal pups were analyzed for methylation and gene expression at P21 (3 weeks), post-weaning.

## Tissue processing and immunohistochemistry

E8.5-E18.5 embryos and their placentas were fixed by overnight immersion in 4% PFA/PBS at 4 °C. Fixed tissues and embryos were dissected and either imaged intact or embedded in paraffin. Paraffin sections were stained with hematoxylin and eosin. Slides were scanned on the 3D Histech Pannoramic midi camera and analyzed using the CaseViewer Digital Slide Viewer. The thickness of the placental layers was calculated as follows: The decidua (Dd) layer was measured as the vertical length from the myometrium to the spongiotrophoblast (junctional zone) layer. The spongiotrophoblast (Sp) layer was measured as the vertical length from the decidua to the labyrinth layer. The labyrinth (Lb) layer was measured as the vertical length from the spongiotrophoblast to the chorionic plate. And the chorionic plate (Cp) layer was measured as the vertical length from the labyrinth to the umbilical cord. These lengths were normalized by the determination of the vertical length of the whole placenta.

## Microscopy and image analysis

Embryo images were captured on a Nikon SMZ18 Stereo Microscope and processed with NIS-Elements D Imaging Software (Nikon), ImageJ, and Adobe Photoshop.

## Bisulfite conversion, PCR, and Sanger sequencing

Bisulfite conversion of DNA was established using the EZ DNA Methylation-Lightning Kit (Zymo Research cat#D5031) following the manufacturer's instructions. The resulting modified DNA was amplified by the first round of nested PCR, following a second round using loci-specific PCR primers (primers are listed in Supplementary Table 3). The first round of nested PCR was done as follows: 94 °C for 4 min; 55 °C for 2 min; 72 °C for 2 min; Repeat steps 1–3 1×; 94 °C for 1 min; 55 °C for 2 min; 72 °C for 2 min; Repeat steps 5–7 35×; 72 °C for 5 min; Hold 12 °C. The second round of PCR was as follows: 95 °C for 4 min; 94 °C for 1 min; 55 °C for 2 min; 72 °C for 2 min; Repeat steps 2–4 35×; 72 °C for 5 min; Hold 12 °C. The resulting amplified products were gel-purified, subcloned into a pGEM®-T Easy cloning vector (Promega cat#A1360), and sequenced.

## PyroSequencing methylation analysis

The procedure and primers for this DNA methylation analysis were described previously (Strogantsev et al. 2015, Sun et al.[50], Kunitomi et al.[51]). In brief, 1 μg DNA was treated using the EZ-96 DNA methylation kit (Zymo Research cat#D5032) in accordance with the manufacturer's instructions. Bisulfite-treated DNA was eluted in 30 μl of elution buffer. Amplicons were generated in a 25 μl reaction volume containing 100 nM forward and reverse primers, 1.25 Units of HotstarTaq DNA Polymerase (Qiagen cat#203203), 0.2 mM dNTPs, and 5 μl of bisulfite-treated DNA. PCR cycle conditions consisted of an initial activation step of 95 °C for 15 minutes, followed by 50 cycles of 94 °C for 30 seconds, specific annealing temperature for 30 seconds, and extension at 72 °C for 30 seconds, followed by a final extension at 72 °C for 10 minutes. Pyrosequencing was carried on PSQ HS96 System using PyroMark Gold Q96 SQA Reagents (Qiagen cat#972812). The degree of methylation at CpG sites (without distinguishing between maternal and paternal alleles) was determined by pyro-Q CpG software.

## Targeted methylation analysis

We used the PBAT capture protocol combining PBAT and hybridization with an RNA probe library (capture), as described in detail in Meir Z. et al. Nature Genetics 2020, https://doi.org/10.1038/s41588-020-0645-y. Bisulfite conversion was performed with the EZ DNA Methylation-Lightning Kit (Zymo Research cat#D5031) following the manufacturer's instructions. Converted DNA samples were subjected to an End repair reaction containing End repair mix and buffer (NEB cat#E6050) and 0.2–150 ng DNA. DNA was purified using 2.5× SPRI Agencourt AMPure XP beads (Beckman Coulter cat#A63881). The eluted product was next subjected to an A-tail reaction including 10 mM dATPs and Klenow Fragment 3′→5′ exo- (NEB cat#M0212). DNA was purified with 2.5× SPRI beads. The clean DNA was tagged with an index oligo adapter in a ligation reaction using the Quick ligase Kit (NEB cat#M2200). Tagged products were then cleaned using 1.3× SPRI beads, and amplified for library preparation with 14 PCR cycles using the KAPA HiFi HotStart Ready Mix kit (Kapa Biosystems cat#KK2601), following the manufacturer's protocol. The reaction mix was then cleaned with 0.7× beads. Final libraries were pooled and sequenced on an Illumina NextSeq system using the 150-bp high-output sequencing kit.

## Reverse transcription of RNA and quantitative real-time PCR

RNA was isolated using the Direct-zol RNA Miniprep Kit (Zymo Research cat#R2052) following the manufacturer's instructions. Reverse transcription was performed on 0.2–1 μg of total RNA using the High-Capacity cDNA Reverse Transcription Kit with RNase Inhibitor including random hexamer primers and the MultiScribe™ Reverse Transcriptase (Applied Biosystems cat#4368814) according to the manufacturer's instructions. All PCR reactions were performed in a 384-well plate on a QuantStudio™ 5 Real-Time PCR System (Applied Biosystems cat#A34322) using Fast SYBR™ Green Master Mix (Applied Biosystems cat#4385610). Relative quantification of gene expression was normalized to the geometrical mean of GAPDH and β-Actin expression levels (primers are listed in Supplementary Table 4) and calculated using the ΔΔCT method, plotted as $2^{-\Delta\Delta CT}$.

## Statistical analysis

At least three biological replicates were performed for all experiments. Jarque-Bera tests were used to determine whether the data has skewness and kurtosis that matches a normal distribution. Statistical differences were determined using a two-tailed unpaired Student's $t$ test and one-way analysis of variance. Data are shown as means with error bars representing the standard deviation. $P$ values of <0.05 were considered significant. Chi-squared tests were performed to determine whether there is a statistically significant difference between the expected and observed genotype frequencies at E8.5, E18.5 and at P21. Pearson's correlation tests were used to statistically determine correlation between IG-CGI$^{\Delta P}$ Embryo weight and Gtl2-DMR methylation at E18.5, Dlk1/Gtl2 expression ratio, and IG-CGI$^{\Delta P}$ Embryo weight at E18.5, Dlk1 and Gtl2 relative expression, Rian and Gtl2 relative expression and Mirg and Gtl2 relative expression in IG-CGI$^{\Delta P}$ embryos.

## Reporting summary

Further information on research design is available in the Nature Research Reporting Summary linked to this article.

# Data availability

The data that support this study are available from the corresponding author upon reasonable request. Sequencing data generated in the course of this study have been deposited in NCBI's Gene Expression Omnibus and are accessible through GEO Series accession number GSE207600. Source data are provided with this paper.

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

## Acknowledgements

We thank the Ferguson-Smith and Stelzer group members for their discussion and advice. We are grateful to Dr. Mayshar for critically reading the manuscript. Y.S. is the incumbent of the Louis and Ida Rich Career Development Chair and is supported by European Research Council (ERC_StG 852865), Moross Integrated Cancer Center, the Israel Cancer Research Fund (ICRF), Helen and Martin Kimmel Stem Cell Institute, Yeda-Sela Center, ISF (1610/18), the Minerva Foundation, Human Frontier Science Program (CDA00023/2019-C) and the Schwartz/Reisman Collaborative Science Program. This research was generously supported by Barry and Janet Lang, Hadar Impact Fund, Lord Sieff of Brimpton Memorial Fund, Janet and Steven Anixter, JoAnne Silva, Maurice and Vivienne Wohl Biology Endowment, and Lester and Edward Anixter Family. Y.S. is a member of the European Molecular Biology Organization (EMBO) Young Investigator Program. A.W.S is supported by the Human Frontier Science Program, the Rothschild Yad Hanadiv Fellowship program, the Humanitarian trust, and the Israel National Postdoctoral Award for Advancing Women in Science. A.F.S. is supported by MRC (MR/R009791/1) and Wellcome Trust (210757/Z/18/Z) grants. Y.S and A.F.S labs were supported by Weizmann UK–Making Connections Collaborative Grant.

## Author contributions

A.W.S, R.B.-Y., A.F.S, and Y.S. conceived and designed the experiments, performed data analysis, and its interpretation. A.W.S, R.B.-Y, N.T, and A.S carried out experiments. M.D assisted with data analyses. C.E. assisted in interpreting the results and input on the manuscript text. A.W.S prepared the figures. A.W.S, R.B.-Y., A.F.S, and Y.S. wrote the manuscript.

## Competing interests

The authors declare no competing interests.
