## [Peer Review File · Nature Communications]

REVIEWER COMMENTS

Reviewer #1 (Remarks to the Author):

Weinberg-Shukron et al present a piece of work examining genotype/epigenotype correlations with measured phenotypes at the imprinted *Dlk1/Dio3* locus in the mouse. The data are centred around the question of whether the imprinted gene expression needs to be derived from a specific parental allele or whether the mechanism is designed to achieve close supervision of gene expression levels in general for a more generic but tight gene dosage control.

The idea of the parental origin being central (illustrated through evolutionary theories around imprinted gene function serving to limit or promote the growth of embryos and placentas according to parental origin) or tight control over dosage, as purported in other studies (for example early microarray expression data that evaluated transcriptional output from the mammalian X chromosome compared to the rest of the genome) being central, is an interesting one.

Here the group have been undertaking studies to further dissect the *Dlk1/Dio3* imprinted locus and have gone on to test the idea that the key is dosage control using a series of detailed deletions of bits of the control region at the *Dlk1/Dio3* locus alongside careful phenotypic measurements.

Evidence for the model proposed comes from previous studies of the region in mouse knock outs plus new results presented here from a new deletion. The group slice up the IG-DMR into 2 regions, one the CGI and the other a distal bit that “was suggested to act as a TRE or transcriptional regulatory element”. The nature of this region is not discussed much, it’s just referenced and labelled. The sentence is a bit wishy washy, perhaps the authors meant for this to be rather vaguely defined? Whatever the case, the authors clearly show that normal development depends on keeping a balanced expression between genes in the *Dlk1/Dio3* locus rather than absolute expression levels. The DNA methylation data and expression data collected are rigorous and well described. They convince the reader that they are appropriately correlated to the collected phenotypic data and lead to the idea that the 3 regulatory elements in the region have a hierarchy and that the parental origin per se, isn’t the leading factor that matters.

The mice tested are IG-CGI WT; IG-CGI Δ Maternal ; IG-CGI Δ Biallelic and IG-CGI Δ Paternal. The fact that viability is a characteristic of the scored crosses is pretty fortunate since it can be scored with confidence. A more subtle effect might have been less convincing. While the results highlight the requirement for balanced gene dosage in normal development, the language around the sentence "...for proper embryonic growth and survival and that is more important than parental origin..." could be re-phrased. They have already illustrated that it's not about importance, but something more mechanistic and factual, this sounds like a pet theory, (which I feel they have argued it really isn't).

Figure 3 is really helpful.

Reviewer #2 (Remarks to the Author):

Weinberg-Shukron, Ben-Yair et al. studied the consequences of deleting a CpG island (CGI) within an intergenic (IG) differentially methylated region (DMR), that is part of the imprinted Dlk1-Dio3 locus.

They generated mice which lacked this CGI on the maternal or paternal chromosome or on both, determined transcript levels, DNA methylation, and phenotypic outcomes (embryo size, placental weight, postnatal survival) and compared their results with previously published findings on the deletion of the entire IG-DMR on maternal or paternal chromosomes.

Moreover, they generated animals with double deletions that lacked the entire IG-DMR on the maternal chromosome and the CGI part of the IG-DMR on the paternal chromosome.

Their main conclusions are:

- i) A dependency between regulatory elements of the Dlk1-Dio3 locus controls parent-specific gene expression.
- ii) Flipping imprinting patterns on parental chromosomes as a consequence of an entire IG-DMR deletion on one allele combined with a CGI deletion on the other allele rescues the lethality of each deletion on its own. They claim (end of the introduction) "our data show that irrespective of the parental origin and epigenetic landscape of the IG-DMR, normal development strictly depends on maintaining a balanced expression between genes of the Dlk1-Dio3 locus, rather than absolute levels."

Comments:

1) Results, page 8:

“Synthesizing the result of the two genetic models shows that normal development cannot occur with biallelic expression of maternal genes and repression of Dlk1 or with biallelic expression of Dlk1 and repression of maternal transcripts (Fig. 3).” ... “This raises the question of whether it is expression from the appropriate parental chromosome or balanced gene dosage per se that is required for postnatal survival.”

The terms “balanced expression, balanced gene dosage” and “absolute levels” are misleading. Their results do not allow to conclude that “absolute levels” are not relevant. To support such statements, mice with equal, but higher or lower levels of all transcripts would need to be studied. Could the lethality of animals with an entire IG-DMR deletion on the maternal chromosome and of animals with a CGI deletion on the paternal chromosome (see Fig. 3) not be due to direct consequences of the loss of expression Gtl2, Rian and Mirg in the former case and of loss of Dlk1 expression in the latter (also taking the mouse strains used in this study into consideration)?

Did the authors cross IG-CGI del/wt females with IG-DMR del/wt males to generate IG-DMR del-P; IG-CGI del-M as a control experiment (results, page 9)?

2) The authors report DNA methylation and gene expression analyses of E8.5 and E18.5 embryos and of postnatal mice. It remains however unclear, which tissues were used for DNA and RNA isolation. Were whole embryos used and which postnatal tissues were analyzed? This is relevant as imprinting of the Dlk1-Dio3 locus is regulated in a tissue-specific manner. Moreover, exact ages of postnatal animals should be given instead of “postnatal pups” (legend Fig. 4), “post-weaning” (Extended Data Fig. 2g).

3) Results, page 9: “Our analysis confirms that the maternal IG-DMR deletion encompassing both the IG-CGI and the IG-TRE leads to a methylated Gtl2-DMR in contrast to maternal deletion of the IG-CGI alone (Figs. 4c, d, and Extended Data Fig. 5d).”

No maternal deletion of the IG-CGI alone is shown in these figures.

4) Results, page 10: “Consistent with results from IG-CGI del-P, we found that ...”. It should be clearly written which mouse model/genotype these statements refer to.

5) Results, page 10: “Gene expression analysis confirmed the reciprocal gene dosage effects between IG-CGI del-P and IG-DMR del-M, with the latter exhibiting elevated expression of Dlk1 ...”

The level of Dlk1 expression elevation and the reference sample should be clearly stated in the text.

6) Southern blot and PCR analyses of single genetically engineered mESC colonies are mentioned in the Methods section, but not shown.

7) Did the authors test their data for a normal distribution before applying Student's t-tests?

Minor:

The color codes in Extended Data Fig. 5d belong to other panels of this figure.

The term "physiological phenotypes" in the abstract and introduction is incomprehensible. Would "phenotype" not suffice?

Reviewer #3 (Remarks to the Author):

This is a potentially interesting study examining the effect of parental origin vs dosage effect on the phenotype regulated by a hierarchy of regulatory elements at the imprinted DIK1-Dio3 locus. Phenotype reads outs included weight, brown fat content, placenta structure and perinatal lethality. The work is of interest to people working in the genomic imprinting and epigenetic field and highlights that gene dosage is of higher importance than parental origin.

The methodology and results are essentially sound (minor comments listed below. The figures are clearly presented and the results well explained

Minor comments

Stats throughout would be more robust if based on ANOVA rather than repeated t-Tests.

Dorsal brown fat appeared to be reduced in volume (Ext Data Fig. 2d). Why not quantified as for placental layers (Ext Data Fig. 2e,f).

Survival rates (Ext Data Fig. 2g): Is there already a paucity of pat and bi IG-CGI deletion animals at e8.5? Why not test for significance using Chi-squared?

How could "alterations in placenta structure and function could account for the

perinatal lethality observed in these mutants”?

“At E8.5, mutant embryos appeared phenotypically indistinguishable from WT embryos (Extended Data Fig. 3a)”. Were any measurements made at this stage?

Ideally, an appropriate statistical test (e.g. Spearman’s) should be employed to evaluate the apparent “correlation between Gtl2-DMR methylation levels and embryo weight, with the three embryos exhibiting >20% methylation also demonstrating increased weights (>900 mg; Fig. 1J)”. Incidentally, I can’t find information on the embryo tissue used to prepare DNA for the methylation analysis.

Similarly, correlations in Figs 2c-f should be tested for statistical significance.

The ‘switching’ experiment is genetically very satisfying but the outcome is not surprising in that restoring gene dosage normalises the phenotype. Subtle differences in the weight of ‘rescued’ males is not fully resolved but suggests the restoration is crude in some aspects (again unsurprising given that two different regulatory region mutations have necessarily been used). Importantly, the experiments do resolve the apparent paradox that the two different mutations each result in perinatal lethality.

Much less convincing is the discussion (last paragraph) that artificially switching alleles tells us anything about the evolutionary pressures that drove evolution of imprinted gene regulation.

Similar ‘allele switching’ experiments published on other imprinted loci (Zbdf2, Glaser et al 2021; Peg3, Bretz et al 2018) could be more fully acknowledged.

Reviewer #1:

Weinberg-Shukron et al present a piece of work examining genotype/epigenotype correlations with measured phenotypes at the imprinted Dlk1/Dio3 locus in the mouse. The data are centred around the question of whether the imprinted gene expression needs to be derived from a specific parental allele or whether the mechanism is designed to achieve close supervision of gene expression levels in general for a more generic but tight gene dosage control.

The idea of the parental origin being central (illustrated through evolutionary theories around imprinted gene function serving to limit or promote the growth of embryos and placentas according to parental origin) or tight control over dosage, as purported in other studies (for example early microarray expression data that evaluated transcriptional output from the mammalian X chromosome compared to the rest of the genome) being central, is an interesting one.

Here the group have been undertaking studies to further dissect the Dlk1/Dio3 imprinted locus and have gone on to test the idea that the key is dosage control using a series of detailed deletions of bits of the control region at the Dlk1/Dio3 locus alongside careful phenotypic measurements.

1. Evidence for the model proposed comes from previous studies of the region in mouse knock outs plus new results presented here from a new deletion. The group slice up the IG-DMR into 2 regions, one the CGI and the other a distal bit that “was suggested to act as a TRE or transcriptional regulatory element”. The nature of this region is not discussed much, it’s just referenced and labelled. The sentence is a bit wishy washy, perhaps the authors meant for this to be rather vaguely defined?

Point well taken. The TRE region was previously defined as an active (marked by H3K27ac) and transcribed (as measured by GRO-seq) distal enhancer element (Danko C. G. et al. *Nat. Methods* 2015). In addition, it was shown that transcription factors, including key pluripotency ones, bind this element in mouse embryonic stem cells (see Luo Z. et al. *Genes Dev* 2016). We have modified the sentence in the introduction to include this information. We share the Reviewers’ view that functional characterization of this regulatory element is still lacking, but we believe it is beyond the scope of the current work.

Whatever the case, the authors clearly show that normal development depends on keeping a balanced expression between genes in the Dlk1/Dio3 locus rather than absolute expression levels. The DNA methylation data and expression data collected are rigorous and well described. They convince the reader that they are appropriately correlated to the collected phenotypic data and lead to the idea that the 3 regulatory elements in the region have a hierarchy and that the parental origin per se, isn’t the leading factor that matters.

The mice tested are IG-CGI WT; IG-CGI Δ Maternal ; IG-CGI Δ Biallelic and IG-CGI Δ Paternal. The fact that viability is a characteristic is of the scored crosses is pretty fortunate since it can be scored with confidence. A more subtle effect might have been less convincing.

2. While the results highlight the requirement for balanced gene dosage in normal development, the language around the sentence “...for proper embryonic growth and survival and that is more important that parental origin...” could be re-phrased. They have already illustrated that it’s not about importance, but something more mechanistic and factual, this sounds like a pet theory, (which I feel they have argued it really isn’t).

This is a good point and we agree with the Reviewer. We have modified the respective sentence accordingly: “Our results, therefore highlight a fundamental requirement for balanced gene dosage in the Dlk1-Dio3 locus for proper embryonic growth and survival **regardless of which parental allele genes are expressed from**”.

We have also changed the following sentence at the end of the introduction to further clarify our conclusions: “Importantly, our data show that irrespective of the parental origin and epigenetic landscape of the IG-DMR, normal development strictly depends on maintaining a balanced expression between genes in the *Dlk1-Dio3* locus, rather than **parent-of-origin specific expression**”.

Figure 3 is really helpful.

Reviewer #2:

*Weinberg-Shukron, Ben-Yair et al. studied the consequences of deleting a CpG island (CGI) within an intergenic (IG) differentially methylated region (DMR), that is part of the imprinted *Dlk1-Dio3* locus. They generated mice which lacked this CGI on the maternal or paternal chromosome or on both, determined transcript levels, DNA methylation, and phenotypic outcomes (embryo size, placental weight, postnatal survival) and compared their results with previously published findings on the deletion of the entire IG-DMR on maternal or paternal chromosomes. Moreover, they generated animals with double deletions that lacked the entire IG-DMR on the maternal chromosome and the CGI part of the IG-DMR on the paternal chromosome.*

Their main conclusions are:

*i) A dependency between regulatory elements of the *Dlk1-Dio3* locus controls parent-specific gene expression.*

*ii) Flipping imprinting patterns on parental chromosomes as a consequence of an entire IG-DMR deletion on one allele combined with a CGI deletion on the other allele rescues the lethality of each deletion on its own. They claim (end of the introduction) “our data show that irrespective of the parental origin and epigenetic landscape of the IG-DMR, normal development strictly depends on maintaining a balanced expression between genes of the *Dlk1-Dio3* locus, rather than absolute levels.”*

Comments:

*1. Results, page 8: “Synthesizing the result of the two genetic models shows that normal development cannot occur with biallelic expression of maternal genes and repression of *Dlk1* or with biallelic expression of *Dlk1* and repression of maternal transcripts (Fig. 3).” ... “This raises the question of whether it is expression from the appropriate parental chromosome or balanced gene dosage per se that is required for postnatal survival.*

*The terms “balanced expression, balanced gene dosage” and “absolute levels” are misleading. Their results do not allow to conclude that “absolute levels” are not relevant. To support such statements, mice with equal, but higher or lower levels of all transcripts would need to be studied. Could the lethality of animals with an entire IG-DMR deletion on the maternal chromosome and of animals with a CGI deletion on the paternal chromosome (see Fig. 3) not be due to direct consequences of the loss of expression *Gtl2*, *Rian* and *Mirg* in the former case and of loss of *Dlk1* expression in the latter (also taking the mouse strains used in this study into consideration)?*

We thank the Reviewer for pointing out this important issue. We agree - our data do not allow concluding that absolute levels are not playing a role in the observed phenotype.

Indeed, the lethality of the IG-DMR maternal or the IG-CGI paternal deletions could be in part due to changes in absolute expression levels of maternal/paternal genes. This has been shown in mice with isolated deletions of these genes. Mice lacking *Dlk1* displayed pre- and post-natal growth retardation

and skeletal abnormalities (Moon *et al.* Molecular and Cellular Biology 2002), and mice expressing a double dose of *Dlk1*, are growth enhanced but fail to thrive in early life (da Rocha *et al.* PLoS Genetics 2009). Similarly, deletion or overexpression of *Gtl2* results in severe growth retardation and perinatal lethality (Takahashi *et al.* Human Molecular Genetics 2009, Zhou *et al.* Development 2010 and Kumamoto *et al.* Human Molecular Genetics 2017). These experiments, as well as ours, are not designed to distinguish whether resulting phenotypes are caused by varying ratios of maternal to paternal genes or absolute gene expression since both are most likely affected by the manipulation. To address this, we modified our phrasing and the last paragraph of the introduction has been changed to: “normal development strictly depends on maintaining a balanced expression between genes in the *Dlk1-Dio3* locus, rather than **parent-of origin-specific expression**”.

We also changed the sentence on page 9 to say “This raises the question of whether it is expression from the appropriate parental chromosome or balanced **expression between genes** *per se* that is required for postnatal survival.”

Finally, we further emphasize in the discussion that the current experimental setting does not allow discriminating if the effects result from changes in absolute gene expression, changes in maternal to paternal gene ratios, or both.

“Did the authors cross IG-CGI del/wt females with IG-DMR del/wt males to generate IG-DMR del-P; IG-CGI del-M as a control experiment (results, page 9)?”

We agree with the Reviewer that this reciprocal cross is an important control experiment and have now included this data in our revised manuscript. As expected, double deletion $IG-CGI^{\Delta M}$ and $IG-DMR^{\Delta P}$ were viable. This new information was now added to the results section, page 9: “The reciprocal cross of $IG-CGI^{\Delta WT}$ females with $IG-DMR^{\Delta WT}$ males resulted in viable offspring of all genotypes including double deletion pups. This is expected since mice harboring these single deletions are viable (Extended Data Fig. 5b. N=8, 10, 6 for $IG-DMR^{\Delta P}$, $IG-CGI^{\Delta M}$ and $IG-DMR^{\Delta P}; IG-CGI^{\Delta M}$ respectively. Data not shown)”. We also added a panel in Extended Data Fig. 5b describing these reciprocal crosses and the viability of the offspring.

*2. “The authors report DNA methylation and gene expression analyses of E8.5 and E18.5 embryos and of postnatal mice. It remains however unclear, which tissues were used for DNA and RNA isolation. Were whole embryos used and which postnatal tissues were analyzed? This is relevant as imprinting of the *Dlk1-Dio3* locus is regulated in a tissue-specific manner”*

We thank the Reviewer for this comment and apologize that this information was missing from our data description. Indeed, as we previously showed (Ferrón *et al.*, 2011 and Stelzer *et al.*, 2016), tissue-specific changes in IG-DMR methylation should be considered. To account for this, we used tails (that exhibit intact imprinting in WT) to analyze bulk DNA and RNA from E18.5 embryos. As early post-implantation embryos still present intact parent-specific methylation at the IG-DMR (Stelzer *et al.*, 2015), at E8.5, we used whole-embryo bulk DNA and RNA for our subsequent analysis. This information was now added to the main text and the Methods section under “Embryo analysis”.

“Moreover, exact ages of postnatal animals should be given instead of “postnatal pups” (legend Fig. 4), “post-weaning” (Extended Data Fig. 2g)”

Similarly, we have added this information of exact postnatal ages in:

- Fig. 4d: “**P21 (3 weeks)**” instead of “postnatal” at the y-axes of the methylation graphs.
- Fig. 4 legend: “**P21 (3 weeks)** postnatal pups” instead of “postnatal”.
- Extended Data Fig. 2g: “**P21 (3 weeks)**” instead of “post-weaning”.

- Methods section under “Embryo analysis”: **“Postnatal pups were analyzed for methylation and gene expression at P21 (3 weeks), post-weaning”**.

3. *“Results, page 9: “Our analysis confirms that the maternal IG-DMR deletion encompassing both the IG-CGI and the IG-TRE leads to a methylated Gtl2-DMR in contrast to maternal deletion of the IG-CGI alone (Figs. 4c, d, and Extended Data Fig. 5d).” No maternal deletion of the IG-CGI alone is shown in these figures”*.

We thank the reviewer for pointing this out. We now refer to the correct figures that include methylation analysis of the IG-CGI^{ΔM} (**Fig. 1g and Extended Data Figs. 3c, 4a**).

4. *“Results, page 10: “Consistent with results from IG-CGI del-P, we found that ...”. It should be clearly written which mouse model/genotype these statements refer to”*.

We corrected the sentence to clarify we are comparing IG-CGI^{ΔP} embryos from the double deletion model (Fig. 4a) to IG-CGI^{ΔP} embryos from the isolated CGI deletion model (Fig. 1g): “Consistent with results from IG-CGI^{ΔP} (**Fig. 1g**), we found that **also in IG-DMR^{ΔM};IG-CGI^{ΔP} embryos and their IG-CGI^{ΔP} littermates** both IG-TRE alleles were unmethylated early in development, and *de novo* methylation accumulated on some paternal alleles between E8.5 and E18.5”.

5. *“Results, page 10: “Gene expression analysis confirmed the reciprocal gene dosage effects between IG-CGI del-P and IG-DMR del-M, with the latter exhibiting elevated expression of Dlk1 ...” The level of Dlk1 expression elevation and the reference sample should be clearly stated in the text”*.

We corrected the sentence according to the Reviewers’ comment: “Gene expression analysis confirmed the reciprocal gene dosage effects between IG-CGI^{ΔP} and IG-DMR^{ΔM}, with the **former exhibiting lower levels of Dlk1 relative to WT littermates (~5 folds) together with elevated levels of all maternal transcripts at this locus (~1.5 folds), while the latter showing the reciprocal effects”**.

6. *“Southern blot and PCR analyses of single genetically engineered mESC colonies are mentioned in the Methods section, but not shown”*.

We thank the reviewer for pointing this out and apologize for the confusion - Southern blots were not performed. This is a typo from a previous version of the manuscript that was unfortunately overlooked. We have, however, provided evidence for PCR and sequencing analysis from genetically modified mESCs in Extended Data Fig. 1a and b. The relevant Methods section under “Generation of IG-CGI floxed mESCs” was now corrected.

7. *Did the authors test their data for a normal distribution before applying Student’s t-tests?*

Indeed, we performed a Jarque-Bera test to determine whether or not our data have skewness and kurtosis that matches a normal distribution. The following datasets were all normally distributed and t-tests were used under normal distribution parameters: embryo weight, embryo length, placenta weight, placenta layers width, methylation levels of IG-CGI, IG-TRE and Gtl2-DMR in all genotypes at E8.5 and E18.5 and expression levels of Dlk1, Gtl2, Rian and Mirg in all genotypes at E8.5 and E18.5. We also added a one-way ANOVA test on all data sets to determine statistical significance between groups (without assuming normal distribution).

These points are now clarified in the Methods section under “Statistical analysis”. We have also provided the raw data together with the Jarque-Bera results, t-test and ANOVA p-values in a “Source data file” and have indicated that this data is available in the “Data availability” section at the end of the manuscript.

8. *“The color codes in Extended Data Fig. 5d belong to other panels of this figure”*.

We thank the reviewer for pointing this out. This error was now corrected.

9. *“The term “physiological phenotypes” in the abstract and introduction is incomprehensible. Would “phenotype” not suffice?”.*

Point well taken. The sentence was changed from “physiological phenotypes” to “phenotypes”.

Reviewer #3:

This is a potentially interesting study examining the effect of parental origin vs dosage effect on the phenotype regulated by a hierarchy of regulatory elements at the imprinted DIK1-Dio3 locus. Phenotype reads outs included weight, brown fat content, placenta structure and perinatal lethality. The work is of interest to people working in the genomic imprinting and epigenetic field and highlights that gene dosage is of higher importance than parental origin. The methodology and results are essentially sound (minor comments listed below. The figures are clearly presented and the results well explained

Minor comments

1. *“Stats throughout would be more robust if based on ANOVA rather than repeated t-Tests”.*

We thank the Reviewer for this suggestion and, accordingly, added a one-way ANOVA test to determine statistical significance between groups. We have implemented this test throughout, now included in relevant panels of embryo weights, embryos length, placenta weights, placenta layers width, methylation levels of IG-CGI, IG-TRE and Gtl2-DMR in all genotypes at E8.5 and E18.5 and expression levels of Dlk1, Gtl2, Rian and Mirg in all genotypes at E8.5 and E18.5

These points are now clarified in the Methods section under “Statistical analysis”. We have also provided the raw data together with the Jarque-Bera results, t-test and ANOVA p-values in a “Source data file” and have indicated that this data is available in the “Data availability” section at the end of the manuscript.

2. *“Dorsal brown fat appeared to be reduced in volume (Ext Data Fig. 2d). Why not quantified as for placental layers (Ext Data Fig. 2e,f)”.*

This is an interesting point. As mentioned in the Results section, page 5, histological analysis of the embryos did not detect any gross cellular or morphological defects. The only difference was a reduction in brown fat. However, it was not completely absent, and the reduction was variable between embryos and tissue sections from different areas. While this is clearly an interesting lead, we believe that follow up experiments to dissect and quantify this phenotype is beyond the scope of the current work. We hope that the Reviewer agrees with this point. We now added text elaborating on the variability in brown fat reduction between embryos and sections.

3. *“Survival rates (Ext Data Fig. 2g): Is there already a paucity of pat and bi IG-CGI deletion animals at e8.5? Why not test for significance using Chi-squared?”.*

We observed similar Mendelian distributions of all genotypes in E8.5, including IG-CGI paternal deletion embryos, which displayed normal morphology. The numbers in Extended Data Fig. 2g represent embryos from 7 E8.5 litters. As suggested by the Reviewer, we performed a Chi-squared test and determined there is no statistically significant difference between the expected frequencies and the observed in each litter. We also performed this test at E18.5 (28 litters) and at P21 (16 litters): while the genotype distribution is still normal at E18.5, at P21 there is a statistically significant skew reflecting our observation that biallelic and paternal IG-CGI deletion embryos rarely survive

postnatally. These points are now clarified in the Methods section under “Statistical analysis”. Raw data and statistical analysis were now provided in a “Source data file”.

0. *“How could “alterations in placenta structure and function could account for the perinatal lethality observed in these mutants”?”*.

Functional defects of the placenta can cause several developmental disorders, such as intrauterine growth retardation, or preterm birth in humans and mice (reviewed in Terry K Morgan, Am J Perinatol, 2016). The labyrinth layer is an essential part of the placenta, where there are a large number of fetal capillaries exchanging nutrients and gases between the fetus and mother. It was previously reported that the *Rtl1* gene at the *Dlk1-Dio3* locus plays a role in the maintenance of the fetal capillaries (Sekita *et al.* Nature Genetics 2008) and that this damage to the fetal capillaries is the cause of fetal lethality and late fetal growth retardation in *Rtl1* KO embryos (Kitazawa *et al.* Genes to Cells 2017). We therefore hypothesize that the observed change in imprinting at the *Dlk1-Dio3* locus induced by IG-CGI deletion could result in abnormal placental function and contribute to the embryonic perinatal lethality. We further elaborate on this point in the discussion section of the revised version of the manuscript.

4. *“At E8.5, mutant embryos appeared phenotypically indistinguishable from WT embryos (Extended Data Fig. 3a)”. Were any measurements made at this stage?”*.

As stated above (please see point 3), we could not observe any gross morphological difference between the genotypes at E8.5 and therefore no measurements were made. The paternal deletion embryos appeared at normal length and morphological developmental stage as their WT littermates, with a well-developed axis, clear headfolds, heart rudiment, and first somites. We now further clarify this in the text and the corresponding figure legends.

5. *“Ideally, an appropriate statistical test (e.g. Spearman’s) should be employed to evaluate the apparent “correlation between Gtl2-DMR methylation levels and embryo weight, with the three embryos exhibiting >20% methylation also demonstrating increased weights (>900 mg; Fig. 1J)”*.

As suggested by the Reviewer, we performed a Pearson’s correlation test to statistically determine the correlation between IG-CGI^{ΔP} embryo weight and Gtl2-DMR methylation at E18.5. Pearson’s correlation coefficient = 0.59523, was added to legend of Fig.1j. The raw data and statistic calculations are available in a “Source data file”.

6. *“Incidentally, I can’t find information on the embryo tissue used to prepare DNA for the methylation analysis”*.

We apologize that this information was missing from our data description. This is indeed an important point, as we previously showed tissue-specific changes in IG-DMR methylation during mouse development (Ferrón *et al.*, 2011 and Stelzer *et al.*, 2016). To account for this, we used tails (that exhibit intact imprinting in WT) to analyze bulk DNA and RNA from E18.5 embryos. As early post-implantation embryos still present intact parent-specific methylation at the IG-DMR (Stelzer *et al.*, 2015), at E8.5, we used whole-embryo bulk DNA and RNA for our subsequent analysis. This information was now added to the main text and the Methods section under “Embryo analysis”.

7. *“Similarly, correlations in Figs 2c-f should be tested for statistical significance”*.

As indicated above (#6), we performed a Pearson’s correlation test to statistically determine correlation between expression of *Dlk1* to *Gtl2*, *Rian* to *Gtl2*, *Mirg* to *Gtl2* in IG-CGI^{ΔP} embryos and between *Dlk1/Gtl2* expression ratio and IG-CGI^{ΔP} Embryo weight at E18.5. This information was added

to the Methods section under “Statistical analysis”, and correlation coefficients were added to corresponding panels in Fig.2.

8. *“Much less convincing is the discussion (last paragraph) that artificially switching alleles tells us anything about the evolutionary pressures that drove evolution of imprinted gene regulation”.*

Point well taken. The last paragraph mainly discusses balanced gene expression as a potential general mechanism shared by many imprinted loci and the role of methylation in ensuring this tight regulation. The sentence speculating on selection pressure was modified.

9. *“Similar ‘allele switching’ experiments published on other imprinted loci (Zbdf2, Glaser et al 2021; Peg3, Bretz et al 2018) could be more fully acknowledged”.*

We thank the Reviewer for pointing this out. These studies are now further discussed in the revised version of the manuscript.

REVIEWERS' COMMENTS

Reviewer #2 (Remarks to the Author):

My concerns have all been addressed in the authors' response and revised version of the manuscript.

Regarding my comment 1. I would however recommend to include the reference to the mouse models mentioned in their response:

"Indeed, the lethality of the IG-DMR maternal or the IG-CGI paternal deletions could be in part due to changes in absolute expression levels of maternal/paternal genes. This has been shown in mice with isolated deletions of these genes. Mice lacking *Dlk1* displayed pre- and post-natal growth retardation and skeletal abnormalities (Moon et al. *Molecular and Cellular Biology* 2002), and mice expressing a double dose of *Dlk1*, are growth enhanced but fail to thrive in early life (da Rocha et al. *PLoS Genetics* 2009). Similarly, deletion or overexpression of *Gtl2* results in severe growth retardation and perinatal lethality (Takahashi et al. *Human Molecular Genetics* 2009, Zhou et al. *Development* 2010 and Kumamoto et al. *Human Molecular Genetics* 2017)."

Reviewer #3 (Remarks to the Author):

All my concerns have been addressed

Reviewer #2:

My concerns have all been addressed in the authors' response and revised version of the manuscript.

Regarding my comment 1. I would however recommend to include the reference to the mouse models mentioned in their response:

*"Indeed, the lethality of the IG-DMR maternal or the IG-CGI paternal deletions could be in part due to changes in absolute expression levels of maternal/paternal genes. This has been shown in mice with isolated deletions of these genes. Mice lacking *Dlk1* displayed pre- and post-natal growth retardation and skeletal abnormalities (Moon et al. *Molecular and Cellular Biology* 2002), and mice expressing a double dose of *Dlk1*, are growth enhanced but fail to thrive in early life (da Rocha et al. *PLoS Genetics* 2009). Similarly, deletion or overexpression of *Gtl2* results in severe growth retardation and perinatal lethality (Takahashi et al. *Human Molecular Genetics* 2009, Zhou et al. *Development* 2010 and Kumamoto et al. *Human Molecular Genetics* 2017)."*

We thank the reviewer for pointing this out and apologize that these references were missing from the manuscript. We have now added these in the discussion, page 12, references #35, and 40-43:

"...genetic models presented in this study do not easily predict previously shown developmental phenotypes associated with the individual knockout of either *Dlk1* or *Gtl2*³⁵⁻⁴³".

Reviewer #3:

All my concerns have been addressed